# The Effects of Pt-Doped TiO_2_ Nanoparticles and Thickness of Semiconducting Layers at Photoanode in the Improved Performance of Dye-Sensitized Solar Cells

**DOI:** 10.3390/ma15227941

**Published:** 2022-11-10

**Authors:** M. Mujahid, Omar A. Al-Hartomy

**Affiliations:** 1Physics Section, SHSSSB, Aligarh Muslim University, Aligarh 202002, India; 2Department of Physics, Faculty of Science, King Abdulaziz University, Jeddah 21589, Saudi Arabia

**Keywords:** power-output, electrochemical impedance, titania, electrical polarization, ethanol

## Abstract

This work synthesized Pt-doped dye-sensitized solar cells (DSSC) with different molar ratios and thicknesses. The materials were revealed fully through X-ray diffraction (XRD), energy-dispersive spectroscopy (EDS), and transmission electron microscopy (TEM). The photovoltaic properties of the sample were studied by UV-visible spectroscopy, electrochemical impedance spectroscopy (EIS), and IPEC (incident photon-to-current conversion efficiency) techniques. EIS analysis established the decrease in series resistance at the electrolyte interface. It could be one of the reasons for the increase in electron transfer rate and decrease in the recombination process at the interface. Statistical data obtained from optical and electrical investigations revealed that the electrical power-output efficiency of DSSC was 14.25%. It was found that a high ratio of Pt doping and thinner thickness can promote cell performance, owing to the reduction of series resistance, lower bandgap, and high dye adsorption. Doping TiO_2_ with Pt reduced its energy bandgap and introduces intermediate energy levels inside TiO_2_ to facilitate the transition of electrons at low excitation energies. The absorbance of the samples 0.15 M Pt and 0.25 M Pt showed improvement in the wavelength ranging from 200 to 800 nm by Pt doping.

## 1. Introduction

The dye-sensitized solar cell (DSSC) is an inexpensive and versatile cell that is based on photovoltaic technology. It contains a semiconductor between a photosensitized anode, dye, liquid electrolyte, separator, and counter electrode. It converts solar light into useful energy. This photovoltaic cell, also called the Grätzel cell, was devised by Brian O’Regan and Michael Grätzel [1].

Titanium dioxide (TiO_2_) is also known as a metal oxide nanoparticle (MO-NPS) [2]. It is chemically stable and non-toxic. [3]. Compared with other MO-NPS such as, for example, Al_2_O_3_, CuO, CO_3_O_4_, Cr_2_O_3_, Fe_3_O_4_, La_2_O_3_, Mn_2_O_3_, NiO, ZnO, and ZrO_2_ with energy bandgaps of 8.30 eV, 1.34 eV, 2.43 eV, 3.08 eV, 1.85 eV, 5.77 eV, 2.99 eV, 3.66 eV, 3.31 eV, and 5.04 eV respectively, the energy bandgap of TiO_2_ is 3.2 eV [4,5,6]. TiO_2_ shows electron diffusivity, large binding energy, and bulk mobility [7,8]. TiO_2_ is used in many electronic devices, such as variable resistors [9,10], piezoelectric devices, and transducers [11,12]. There are several methods to synthesize TiO_2_, such as co-precipitation, ball-milling, sol–gel, and the hydrothermal method [13,14].

TiO_2_ is UV-active, and doping TiO_2_ with Pt supports its photocatalytic behavior toward visible light. Pt is a valuable noble metal and a perovskite booster [15]. Doping TiO_2_ with Pt reduces its energy bandgap and introduces intermediate energy levels inside TiO_2_ to facilitate the transition of electrons at low excitation energies.

This publication’s novelty is to improve solar cell efficiency; with this objective, we tailored and processed the synthesis of Pt-doped DSSC with different molar ratios and thicknesses. We found that a high molar ratio of Pt doping and thinner thickness can promote cell performance. This work will be advantageous for studying indoor and outdoor panel displays, glass tiles, and many nanostructure-based devices [16,17,18].

Many researchers have investigated the effects of doping and the thickness of photoandes on synthesizing DSSC. In 2004, Meyer et al. used bis-tetra-butylammonium (N719 dye) and calculated the output efficiency of the cell as 11.2% [19]. In 2014, Korake et al. published research using a naphthyridine complex compound and found a maximum power conversion efficiency of 7% [20]. In 1998, Lieber et al. determined the effect of the TiO_2_ nanospindle photoanodes and achieved an efficiency of 8.30% [21]. In 2020, Chih-Hung et al. used triethylamine to form Cu_2_ Zn Sn S_4_ nanoparticles to fabricate solar cells [22]. In 2022, Chen et al. used nano-ferrite zinc-rich materials to fabricate DSSC [23]. Thus, to improve cell performance, the new materials were synthesized and applied to the photo-anode of the DSSC. To date, the enhancement in cell efficiency was recorded to be 33.8% using FeT-based samples [24].

Indeed, some electrical characterization is necessary to flag and fix the effortless charge transport and charge separation characteristics for passive and possible maintenance of the output power efficiency of the cell.

## 2. Experimental Techniques

### 2.1. Chemicals and Reagents

3-Chloroperbenzoic acid (3-CPBA), titanium IV oxysulfate hydrate, platinum nitrate, ammonium hexachloroplatinate, and bis-tetra-butylammonium (N719) dye) were procured from Sigma-Aldrich Company, St. Louis, MO, USA. Surfactant Triton X-100, benzene, 2-propanol, ethyl alcohol, acetic acid, and sulphuric acid were procured from Merck Company, St. Louis, NJ, USA. All the reagents used were of analytical grade.

### 2.2. Hydrothermal Processing

The hydrothermal process is a cheaper synthesis process to obtain nanoparticles of uniform shapes and sizes. Oxides, sulfide, carbonate, and tungsten nanoparticles can be easily synthesized using this process.

In the hydrothermal process, chemical precursors were dissolved in water in an autoclave containing temperature and pressure control gauges. To start this process, a mixture of 9 g of titanium IV oxysulfate hydrate powder was stirred inside the autoclave at 180 °C for 24 h with 5 g of 3-CPBA in 200 mL of benzene followed by centrifugation for one hour. The solid material collected after centrifugation was washed 3–4 times with deionized water and ethyl alcohol. Finally, the collected materials were dried at 90 °C for 7 h.

A total of 3 g of surfactant Triton X-100 prepared in 500 mL of deionized water was regularly stirred for 10 h in an oven containing 2 g of titanium IV oxysulfate hydrate prepared in 50 mL of isopropyl alcohol. Following this, both solutions were mixed. Then, 50 mL of concentrated sulfuric acid was added dropwise into the solution under continuous magnetic stirring at 50 °C for 10 h. The synthesized materials underwent hydrothermal treatment at 180 °C for 24 h. The mixture was finally cooled at room temperature, and the collected solid materials were filtered and washed with distilled water and ethanol, followed by air-drying. The sample was then calcinated at 400 °C for 16 h in a nitrogen atmosphere. For doping of TiO_2_ with Pt, different molarity concentrations of platinum nitrate (0.15 M Pt, 0.25 M Pt, 0.35 M Pt, and 0.45 M Pt) were added dropwise into the surfactant solution.

### 2.3. Layering Procedure on the Photo Anodes and Their Control

The vital role of synthesizing the cell was to prepare the paste. To start with, the synthesized material was mixed with 0.035 M dilute acetic acid and 2–3 mL of Triton X 100 to prepare a paste. The film thickness was controlled by the number of tapes used for the doctor blade method [25]. The prepared paste was coated on the FTO glass. The glass plate was warmed at 450 °C with the help of electric hot plates for 40 min and allowed to cool at room temperature. Three different thicknesses of semiconducting layers, namely, 12 μm, 16 μm, and 20 μm, were prepared following this method by controlling time, pasted material, and temperature. The surface profiler was used to measure the film thickness.

Each FTO glass plate prepared by the above method was dipped in a beaker containing 0.5 mM ethanol solution and N719 dye for 8 h. To boost electron recombination and charge transfer, the working electrode of the cell was coated with platinum. The electrolyte with iodine content was introduced into the cell. Hot melt foil seal cathodic and anodic plates were used to protect the cell from short-circuiting, leakage, and solvent evaporation.

### 2.4. Electrochemical Impedance Spectroscopy (EIS)-Nyquist Curve

The EIS–Nyquist curve was plotted with the help of software. Accordingly, precise values of fitted resistance, capacitance, and constant phase element were obtained.

### 2.5. Characterization Equipment

The prepared samples were characterized by an XRD-having wavelength λ = 0.15418 nm. A UV–VIS spectrophotometer, Jasco, OR, USA, modal number V-770, was used to analyze the absorption spectrum. TEM carried out shapes, sizes, and particle size distribution under different magnifications. A semi-automated Keithley (British Columbian Company) meter modal 2611A was used to obtain the current-voltage characteristics curve. The IPEC spectrum was measured by a multimeter, model number PEC S-20. EIS measurement was carried out using an impedance analyzer in the frequency range of 20 Hz to 1 MHz. The stylus profiler measured the sample thickness.

Figure 1 depicts the schematic representation of the cell containing Pt-doped TiO_2_ coated on the FTO conductive glass plate sensitized with dye. A separator separated the anode and cathode. The liquid electrolyte of iodide/tri-iodide redox couple was injected through the hole drilled towards the counter electrode. The cathode also consisted of an FTO conductive glass plate with a deposited platinum layer. When light was incident on the photoanode of the cell, electrons moved from the valance band to the conduction band. Hence, a voltage was set up across the junction.

## 3. Analysis of Results

### 3.1. Analysis of the XRD Result

Figure 2 represents an XRD graph of Pt-doped and undoped TiO_2_ samples with different molar ratios. Comparing the XRD graph with JCPDS card number 21-1272, the phase of TiO_2_ was indexed as the anatase phase [25]. The XRD graph consisted of four diffraction peaks at 25.2, 38.9, 48.0, and 53.0 at angle 2theta (degrees). These peaks corresponded to diffraction planes of (101), (004), (200), and (105), respectively. No characteristic peaks of other defects were observed. These results indicated that the products had crystallinity and purity.

The diffraction peaks of Pt-doped TiO_2_ NPs were in good agreement with those of the anatase phase of the TiO_2_ and were almost identical to those of undoped TiO_2_, indicating that the Pt atoms were doped into the TiO_2_ crystal. With the increase in Pt doping concentration, the diffraction peaks remained at the same diffraction angle and the exact location; this illustrated that the anatase crystalline phase of TiO_2_ remained unchanged with Pt doping. The peak intensity increased with increasing doping concentrations. This was due to the expansion of the unit cell, which was due to the presence of Pt ions at the interstitial site of TiO_2_. Our XRD result was in good agreement with the reported result of Rosario and Pereira [26].

The full width at half maxima (FWHM) of the diffraction peak (101) was used to calculate crystallite size (d_hkl_). The Scherrer formula [27] was used to calculate the crystallite size of the diffraction peak (101), and its value was found to be 20 nm.
d_hkl_ = 0.9λ/βcosƟ(1)
where Ɵ is the diffraction angle, β is FWHM, and λ is the wavelength of light used.

### 3.2. UV-Visible Spectra

Figure 3 shows the UV-visible spectrum of TiO_2_ samples doped with the distinct molar ratio of Pt. The absorbance of the samples 0.15 M Pt and 0.25 M Pt showed improvement in the wavelength ranging from 200 to 800 nm by Pt doping. Although there was no improvement in absorbance in the samples doped with 0.35 M Pt and 0.45 M Pt, we may speculate some organic molecules, such as surfactants, may have modified the pattern. However, other samples doped with 0.15 M Pt and 0.25 M Pt showed clear improvement in absorbance compared to undoped samples.

It is also clear from Figure 3 that after doping, the absorption edge shifted from 360 nm to 460 nm. This shift was ascribed to the reduction of the bandgap. This observation was the same as the reported value of Chunqiao et al. in the case of doping of TiO_2_ by La and Eu [27]. The bandgap energies (E_g_) of synthesized material were found using the Tauc formula given by Equation (2):(αhν)^2^ = hν − E_g_(2)

When the value of (αhν)^2^ was plotted on the y-axis, and photon energy E (eV) was plotted on the x-axis, then the pattern of this graph was as shown in Figure 4.

If we compare Equation (2) with linear equation y = mx − c, by putting y = 0, Equation (2) can be reduced to
E_g_ = hν

When the linear region of the curve drawn in Figure 4 is additionally plotted, their intersection on the x-axis will provide bandgap energy. The calculated Eg values for undoped and doped samples using the calculation shown Figure 4 were found to be 3.2 eV (undoped TiO_2_), 3.02 eV (TiO_2_ doped with 0.15 M Pt), 2.9 eV (TiO_2_ doped with 0.25 M Pt), 2.84 eV (TiO_2_ doped with 0.35 M Pt), and 2.75 eV (TiO_2_ doped with 0.45 M Pt). The energy bandgap values of Pt-doped samples were observed to be diminutive in energy than undoped samples. Due to this reduction in energy bands, Pt-doped TiO_2_ material showed an increase in photovoltaic action under sunlight irradiation [28]. The bandgap reduction was due to the formation of substitutional defects by incorporating Pt ions into the TiO_2_ crystal [29]. Defects, in general, are defined as those in which there are irregularities in the arrangements of constituent particles.

### 3.3. TEM Analysis

Figure 5 shows the TEM image of one of the samples doped with 0.25 M of Pt onto TiO_2_ composite material. Figure 5 also indicates that the mean dimension of synthesized TiO_2_ nanoparticles was 20 nm. Thus, results obtained from TEM appear to be compatible with the results obtained using the Scherrer formula. These two methods provide good information about synthesized NP size and morphology. We also found dark-colored particles overlapping TiO_2_ particles. They seemed agglomerated because of the complexing agent used during the sample preparation.

### 3.4. EDS Spectra

Figure 6 represents the energy dispersive spectroscopy (EDS) spectra of the synthesized material. EDS provides information about qualitative and semi-quantitative data analysis [30]. The spectra showed three different peaks of Ti, O, and Pt. The peak intensity of Pt was very small. This confirmed the doping of Pt on TiO_2_. Additionally, EDS spectra quantitatively confirmed the weight and atomic weight (%) of Ti, O, and Pt.

### 3.5. Electrical Properties

#### 3.5.1. Performance of DSSC

The photovoltaic performance was studied using a solar photovoltaic radiator AM 1.5 G of intensity 100 mW/cm^2^. The characteristic curve for undoped and doped material is plotted in Figure 7. Equations (3) and (4) were used to calculate the electrical properties of DSSC and summarized in Table 1.

The equation estimated the efficiency of the DSSC.
η% = FF × Jsc × Voc/Pi(3)

The fill factor (FF) was calculated by the following equation:FF = J_max_ × V_max_/J_sc_ × V_oc_.(4)
where J_max_ and V_max_ represent maximum current density and maximum output potential. The Keithley meter was used to obtain a current–voltage characteristic curve. It was clear from the current–voltage characteristic curve that the short circuit current increased with doping concentration. Thus, with increasing doping concentration, the current density, open circuit voltage, fill factor, and efficiency of the cell increased.

#### 3.5.2. Electrochemical Impedance Spectroscopy

Electrochemical impedance spectroscopy (EIS) is used to analyze Lissajous figures. Electrochemical impedance is the electrical parameters of the circuit. Impedance is measured by applying a small signal. EIS can be used to study the charge transfer and diffusion process. The impedance of the cell is described as the sum of real and imaginary parts given by
Z*(ω) = Z′(ω) + jZ′′(ω)(5)

If the real part is plotted on the x-axis and the imaginary negative part on the y-axis, we obtain a plot called the Nyquist plot. Figure 8 shows the Nyquist plot of our synthesized solar cell. This plot informs us about charge transfer from the solution to the electrode surface. The intercept of the semi-circles on the x-axis near the origin represents the cell’s series resistance (R_s_). This plot was conducted at high frequency. Accordingly, the intercept of the consecutive plots near the origin were 4 Ω, 5 Ω, 6 Ω, 7 Ω, and 11 Ω corresponding to samples doped with 0.45 M Pt, 0.35 M Pt, 0.25 M Pt, and 0.15 M Pt, as well as the undoped sample, respectively. Thus, with increasing doping concentration, the series resistance of the cell decreased. Our findings exactly match the results obtained by Yang et al. in the case of TiO_2_ nanotubes treated with TiCl_4_ [30,31,32]. Thus, we found a very important intrinsic material property: doping of the sample at high molar concentration favored the ability of the DSSC to transfer an electron from the solution to the electrode surface, influencing conductivity, current density, and efficiency.

We also found a straight line corresponding to 0.35 M Pt. The intersection of this line on the x-axis provided us with a resistance that not only depended on frequency but corresponded to the diffusion of redox species at the electrode–electrolyte interface. This resistance is called the Warburg resistance [33,34,35].

When each feature of the EIS was characterized by an electrical circuit using software that contains resistance, capacitance, or constant phase elements that are connected in series or parallel to form an equivalent circuit, as shown in Figure 9, then we found that the resistance R_1_ was in parallel with the constant phase element CPE_1_, which was responsible for impedance at the Pt/electrolyte interface [36,37]. The CPE distributed element was introduced for dispersive characteristics of the interfacial capacitance [38,39]. The resistance R_2_, in parallel with the constant phase element CPE_2_, is associated with impedance between the glass substrate and TiO_2_ film [40]. The best-fitted values for the equivalent circuit were given by
R_s_ = 4 Ω, R_1_ = 500 Ω, CPE_1_ = 100 pF, R_2_ = 100 Ω, CPE_2_ = 300 pF.

#### 3.5.3. Incident Photon-to-Current Conversion Efficiency (IPCE) Spectra of TiO_2_-Based DSSC

The IPCE spectrum was plotted in the 350-800 nm wavelength range for the DSSC prepared with undoped and Pt-doped TiO_2_ nanocomposite, and it is depicted in Figure 10.

IPCE (%) is the ratio of the intensity of the output photon to that of the incident photon.
IPCE (%) = 1240 × J_sc_/λP_i_

Figure 10 shows that our prepared DSSC worked in the visible region of the solar spectrum by transferring the energy from dye to TiO_2_ through Pt, which increased the overall photocatalytic activity. The maximum IPCE peak of 80% occurred at a wavelength of 550 nm [41,42]. When the IPCE spectra of the undoped versus Pt-doped sample were compared, we found that the sample doped with a high concentration of Pt had the highest IPCE values compared to the undoped sample. The IPCE value for 0.45 M Pt was 78%, while that of the undoped sample was only 60%. This means that the sample doped with 0.45 M Pt was most effective in collecting light radiation. After absorption through N719 dye, it quickly transferred an electron to the counter electrode than the other doped samples. The net contribution can be viewed as a lower bandgap, reduction of series resistance, a high value of short circuit current density, open circuit voltage capabilities, and high dye adsorption favoring the improvement of performance of DSSC. Thus, we found that the 0.45 M Pt-doped sample was the best suited for photoanode material and therefore promotes cell performance.

#### 3.5.4. The Adequate Thickness of Semiconducting Layers at the Photoanode

Three different semiconductor layers were synthesized, namely, 12 μm, 16 μm, and 20 μm during the synthesis process, which was used as a photoanode. The thickness of the film was measured using a surface profiler.

The current–voltage characteristic curve of Figure 11 shows that the short circuit current increased with the decrease in thickness. The value of current density was found to be 13 mA/cm^2^ for a sample thickness of 12 μm (at a higher doping ratio), 12 mA/cm^2^ for a sample thickness of 16 μm (at an average ratio), and 10.1 mA/cm^2^ for a sample thickness of 20 μm (at an average ratio). Thus, thinner thickness and high ratio can promote cell performance. This high current density was due to the high adsorption of N719 dye. Thus, N719 dye had a higher stability than Ru-based N749 and N3 dye [43,44,45,46].

Figure 12 depicts the IPCE spectrum in the wavelength range 350–800 nm for the DSSC prepared with different thicknesses of photoanode. The maximum IPCE peak value of 75% occurred at a wavelength of 520 nm [47,48,49]. The IPCE (%) value was much higher for the sample with a thinner thickness and high molar ratios, which revealed that the sample was more efficient in collecting photo-excited electrons.

## 4. Conclusions

In this manuscript, we report the synthesis of TiO_2_ nanoparticles using different Pt concentrations varying from 0.15 M Pt to 0.45 M Pt. Various optical and electrical methods fully characterized the prepared substances. We found that the high molar ratios of Pt doping and thinner thickness of semiconducting layers at the anode significantly contributed to promoting cell performance. Three different semiconductor layers were synthesized, namely, 12 μm, 16 μm, and 20 μm during the synthesis process. The thickness of the film was measured using a surface profiler. The high current density of the cell was due to the high adsorption of N719 dye. The current–voltage characteristic curve revealed that the short circuit current increased with the decrease in thickness and high ratio of Pt doping. The thinner thickness and a high ratio of doping can promote cell performance. Statistical data obtained from optical and electrical investigations revealed that the electrical power–output efficiency of our synthesized DSSC was 14.25%. EIS analysis established the decrease in internal resistance at the electrolyte interface. It could be one of the reasons for the increase in electron transfer rate and decrease in the recombination process at the interface. DSSC has numerous applications in panel display devices and many more optoelectronic devices.

## Figures and Tables

**Figure 1 materials-15-07941-f001:**
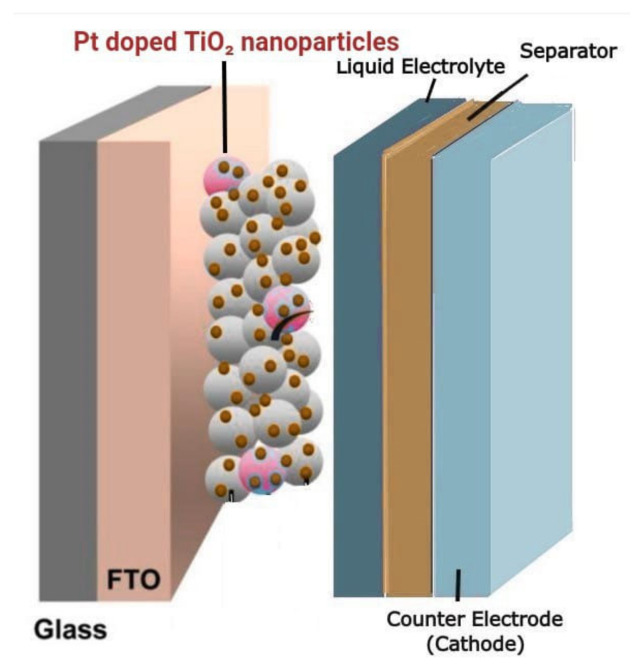
Schematic of TiO_2_-based DSSC.

**Figure 2 materials-15-07941-f002:**
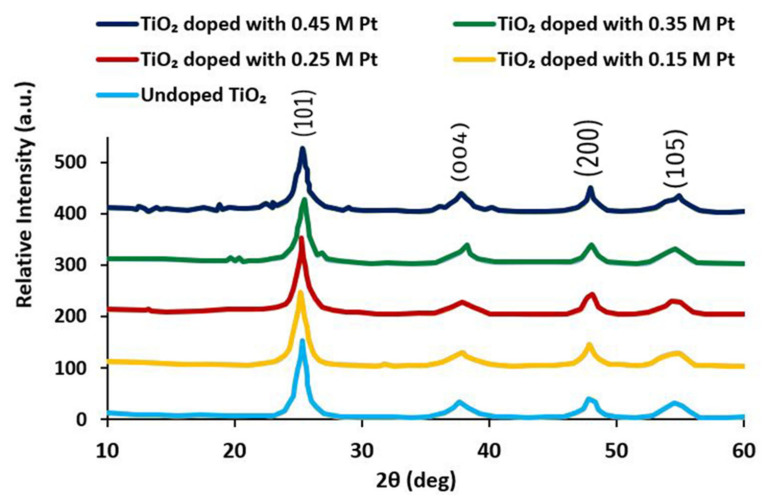
XRD patterns of undoped TiO_2_ and Pt-doped TiO_2_ at different molar concentrations; calcination temp: 400 °C, calcination time: 16 h.

**Figure 3 materials-15-07941-f003:**
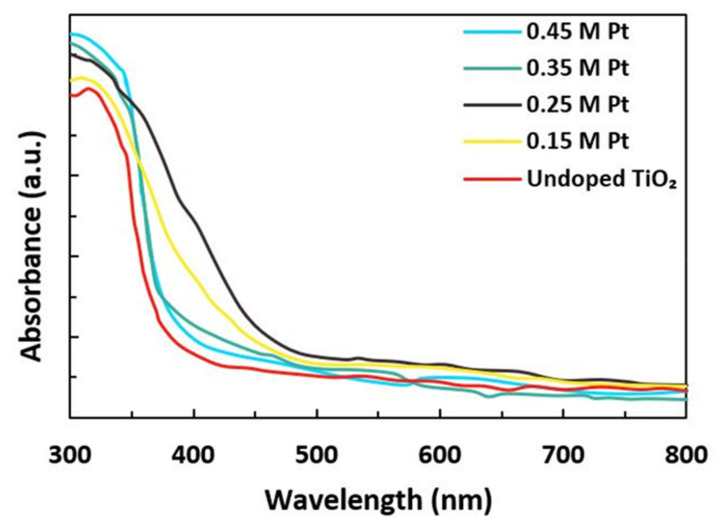
Absorption spectra of undoped TiO_2_ and Pt-doped TiO_2_ at different molar concentrations.

**Figure 4 materials-15-07941-f004:**
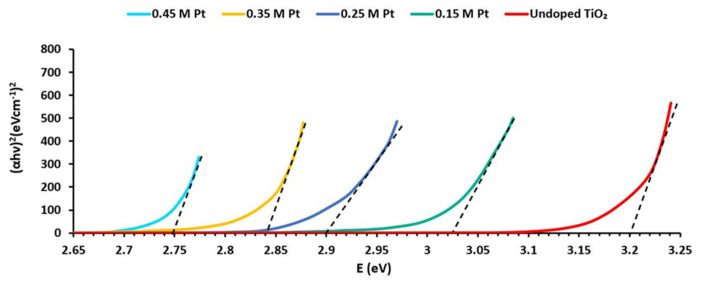
Band gap calculation using the Tauc method.

**Figure 5 materials-15-07941-f005:**
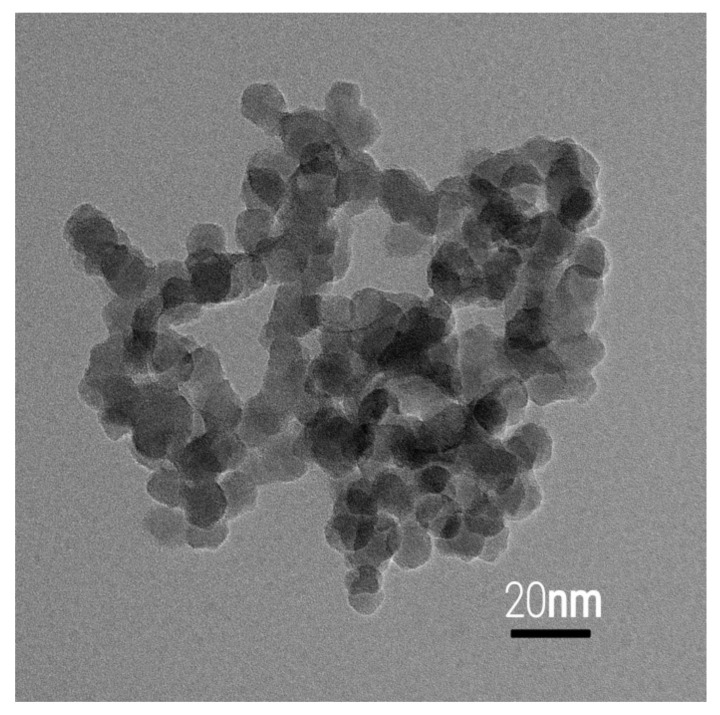
TEM images of Pt-doped TiO_2_.

**Figure 6 materials-15-07941-f006:**
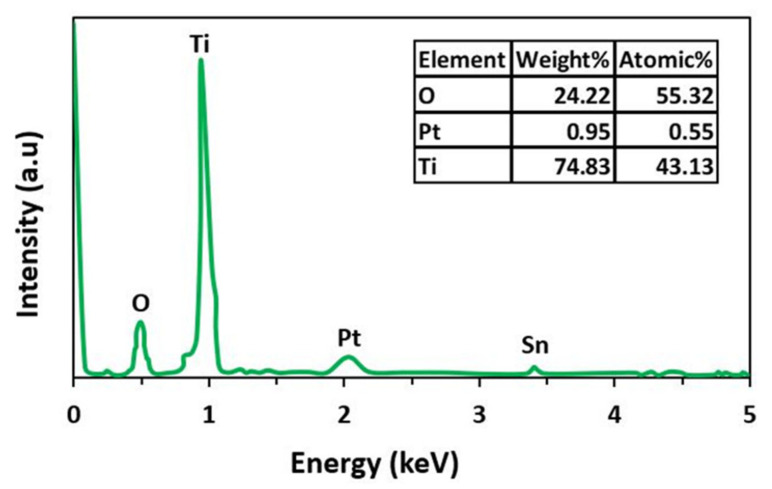
EDS spectra of nanocomposite material.

**Figure 7 materials-15-07941-f007:**
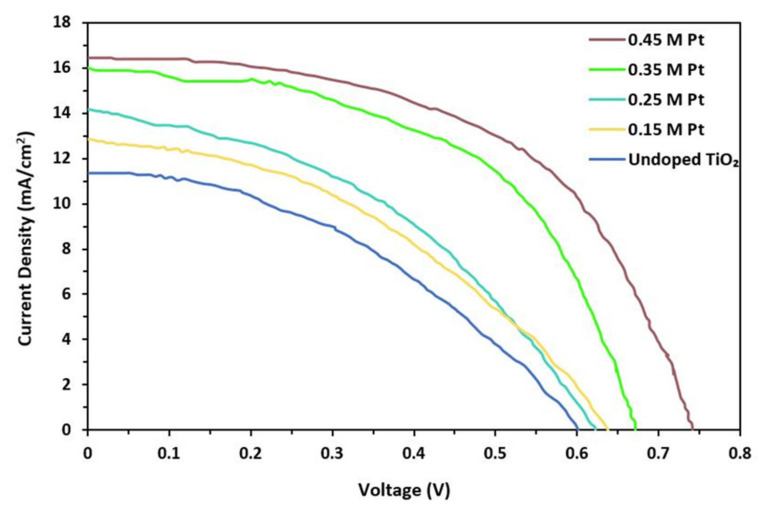
Current density versus photovoltage of undoped TiO_2_ and Pt-doped TiO_2_ at different molar concentrations.

**Figure 8 materials-15-07941-f008:**
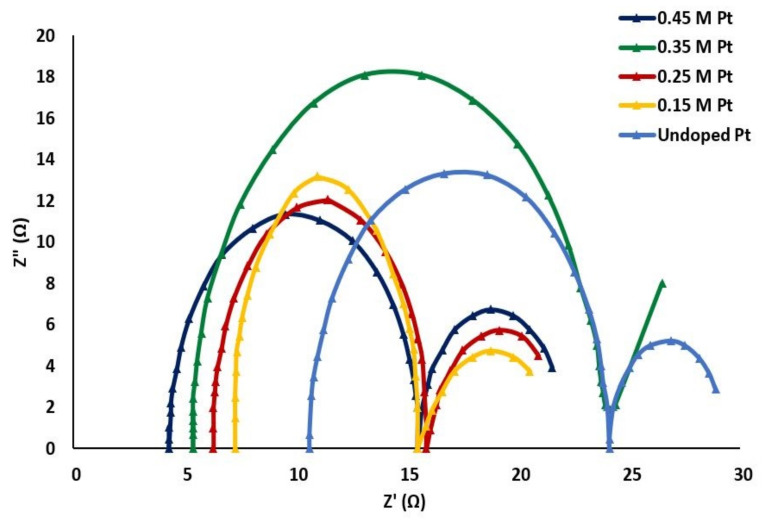
Nyquist plot of undoped TiO_2_ and Pt-doped TiO_2_ at different molar concentrations.

**Figure 9 materials-15-07941-f009:**
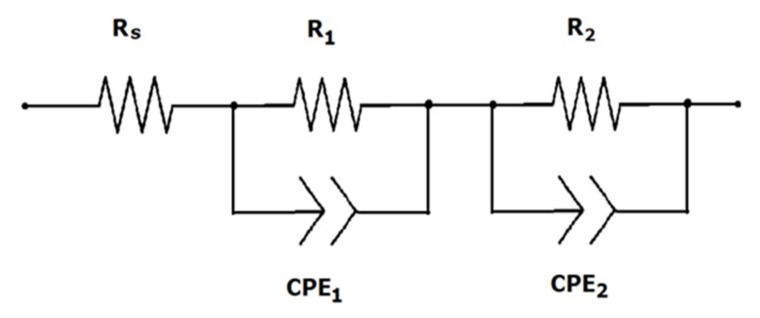
Equivalent circuit of the DSSC.

**Figure 10 materials-15-07941-f010:**
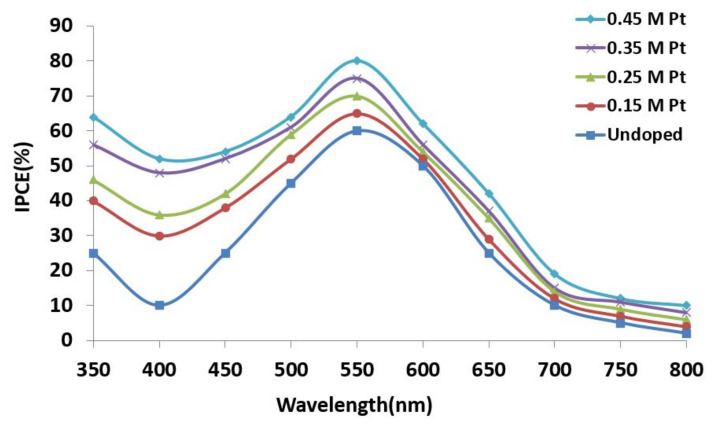
IPCE spectra of undoped and Pt-doped TiO_2_ at different molar concentrations.

**Figure 11 materials-15-07941-f011:**
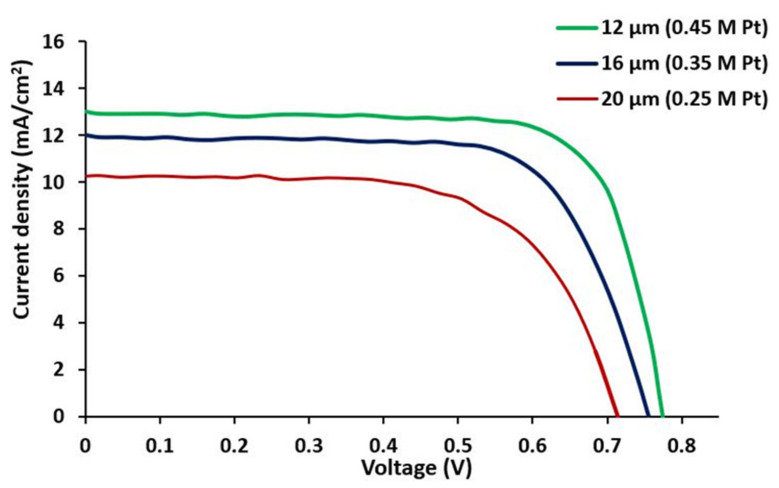
Current density versus photovoltage at three different thicknesses and molar ratios.

**Figure 12 materials-15-07941-f012:**
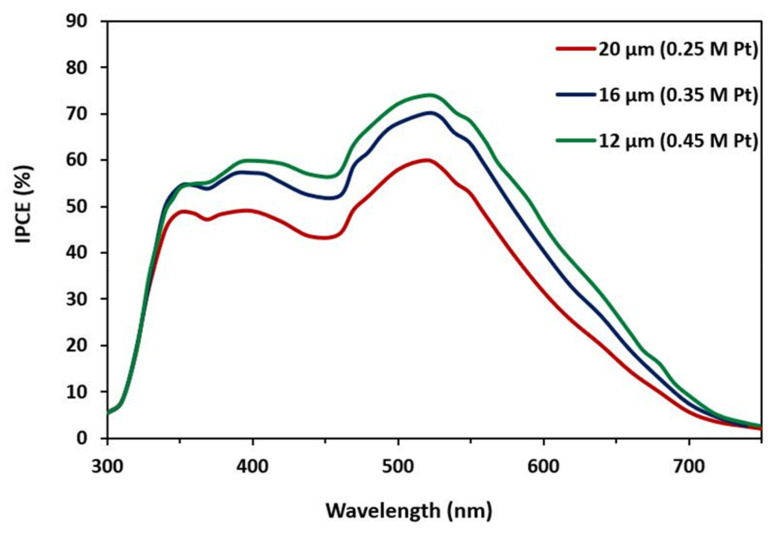
IPCE spectra at three different thicknesses and molar ratios.

**Table 1 materials-15-07941-t001:** Electrical and optical parameters of DSSC.

% Pt Content (M)	IPEC Peak (%)	J_SC_ (mA/cm^2^)	(V_oc_) (volt)	F	Efficiency (%)
Undoped **TiO_2_**	60	11	0.75	0.54	1.45
**TiO_2_ doped with 0.15M Pt**	70	12.5	0.72	0.59	4.25
**TiO_2_ doped with 0.25 M Pt**	72	13.1	0.86	0.69	5.22
**TiO_2_ doped with 0.35 M Pt**	75	16.5	0.92	0.79	11.92
**TiO_2_ doped with 0.45 M Pt**	78	17.5	0.96	0.89	14.25

## Data Availability

The data supporting this study’s findings were available from the corresponding author.

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
