# Peer review of "The Effects of Pt-Doped TiO2 Nanoparticles and Thickness of Semiconducting Layers at Photoanode in the Improved Performance of Dye-Sensitized Solar Cells"

_materials, 2022, doi:10.3390/ma15227941_

Round 1

Reviewer 1 Report

Please refer to the word file. 

Reviewer 2 Report

The manuscript titled " The effects of Pt-doped TiO2 nanoparticles and thickness of semiconducting layers at photoanode in the improved performance of dye-sensitized solar cells"

In this manuscript, authors reported that TiO2 was synthesized using different Pt doping concentrations (0.15 M-0.45 M). The prepared catalyst was fully characterized by different characterization techniques such as UV-visible spectroscopy, X-ray diffraction (XRD), energy-dispersive spectroscopy (EDX), transmission electron microscopy (TEM), and electrochemical-impedance spectroscopy (EIS), and incident photon-to-current conversion efficiency (IPCE) techniques. The accomplished samples performance is impressive. Therefore, I would like to recommend published this work after addressing the following points:

1. There are many abbreviations in the text that are not explained when they first appear such as EIS, N 719, ....etc.

2. Introduction is well-organized and well-written, but the importance and novelty of the research should be highlighted and more clearly stated. The authors give some examples of works in the bibliography, but which is the advantage of their work in comparison with those works.

3. The authors are responsible for the English, which should be polished throughout the manuscript to clear some minor typo/grammar errors.

4. In the introduction part, Some publications are suggested to refer to improve the quality of the manuscript, such as:

 https://doi.org/10.1007/s10904-022-02389-8

https://doi.org/10.1016/j.heliyon.2022.e09652

https://doi.org/10.1016/j.jobe.2022.104869

5. All equation should be revised, which contain some typo error.

6. The author should better improve the beauty and quality of the figures in the manuscript.

Reviewer 3 Report

results representation and discussion part should be further improved. please refer to the comments.

Round 2

Reviewer 1 Report

Please see attached file. Thank you.

Author Response

Please see the attachmet

Reviewer 2 Report

Accepted in the present form

Reviewer 3 Report

Page 2, line 54-56. This sentence is very hard to understand. It is still not clear what is the advantage of this work;

Page 3, line 141. It isn't called impurity. It should be defect.

Page 4, line 153. Adraine V. and Ernesto C? add reference here.

Page 4, line 164. There is no improvement of 0.35M and 0.45M samples between 360-460 nm. They are very close to the undoped one. Some explanation is needed.

Page 5, line 216-223. Now that table 1 already summarizes all information here, why to include this paragraph? The representation is too messy and unreadable.

Page 6, line 244. This paragraph is too weird. First, it should be in the method part. Secondly, it is very unprofessional to use such language in academic writing. 

Page 10, Figure 5(b). This is not EDX. There is no obvious rings or dots. It doesn't show any crystalline information. This image shouldn't be included.

Page 11, Figure 7. The title includes (A)(B)(C)(D), but in the image, there is no corresponding legend. And it is unclear where does this come from "(A) ZnO (B) ZnO+0.46 M Pd (C) ZnO+0.46 M Pd +N719 dye (D) ZnO+0.46 M Pd + Natural dye"?

Page 12, Figure 8. The title indicates there are five samples. But in the image, there are only three.

Page 13, Figure 10. Please make the all figure representations have consistent format. It is very weird to suddenly use an excel plot. 

Page 14, Figure 11. It is very suspicious how to drop a curve (0.25M) with just three points (12 um, 16 um, 20 um). With the thickest sample only has 20 um, it is hard to understand why this figure include 20 to 30 um range in x - axis. The image looks like fabricated.
